# It's Over There: Designing an Intelligent Virtual Agent That Can Point Accurately into the Real World

Fan Wu[1*]  Qian Zhou[1,2†]  Ian Stavness[3‡]  Sidney Fels[1§]

[1]University of British Columbia
[2]Autodesk Research
[3]University of Saskatchewan

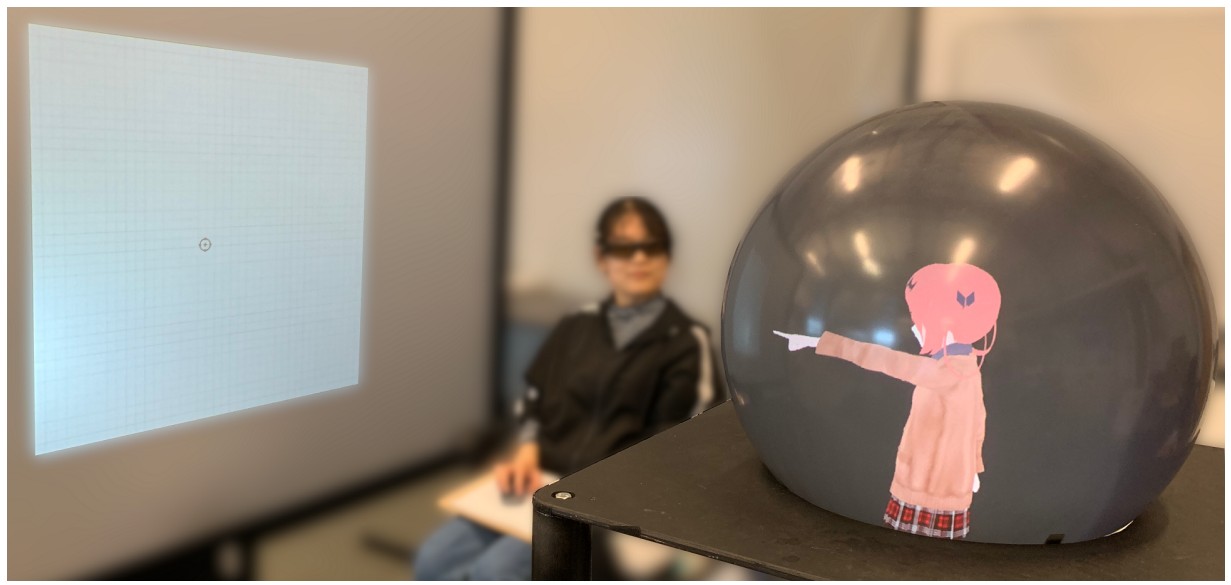

Figure 1: We investigated how accurately users can perceive where an Intelligent Virtual Agent (IVA) rendered in a 3D display is pointing in the real world.

## ABSTRACT

It is challenging to design an intelligent virtual agent (IVA) that can point from the virtual to the real world and have users accurately recognize where it is pointing due to differences in perceptual cues between the two spaces. We designed an IVA with factors including: a situated display, appearance, and pointing gesture strategy to establish whether it is possible to have an IVA point accurately into the real world. With a real person pointing as a baseline, we performed an empirical study using our designed IVA and demonstrated that participants perceived the IVA's pointing to a physical location with comparable accuracy to a real person baseline. Specifically, we found that when the IVA is 230 cm away from the targets on average, the IVA outperformed the real person in the vertical dimension (10.22 cm, 28.8% less error) and achieved the same level of accuracy (11.58 cm) horizontally. Our integrated design choices provide a foundation for design factors to consider when designing IVAs for pointing and pave the way for future studies and systems in providing accurate pointing perception.

**Index Terms:** Human-centered computing—Human computer interaction (HCI)—Interaction techniques—Pointing; Human-centered computing—Interaction design—Empirical studies in interaction design;

*e-mail: wufan95@ece.ubc.ca
†e-mail: qian.zhou@autodesk.com
‡e-mail: ian.stavness@usask.ca
§e-mail: ssfels@ece.ubc.ca

## 1 INTRODUCTION

Many researchers have studied natural human communication cues such as voice and hand gestures [8, 21, 42]. One important aspect of human communication is deictic pointing [18, 52], a hand gesture that complements or replaces verbal communication indicating a point of interest in a shared environment [36]. The pioneering work, *Put that there* [9] demonstrated how an intelligent virtual agent (IVA) can recognize and interpret a person's pointing gestures at objects in a 2D virtual world to facilitate natural human-computer interaction. This work motivates the reverse question, "Can an IVA point back to the real world?" More recently, with the advances in voice-based IVAs, such as Amazon Alexa, the emerging 3D display technologies provide opportunities for IVAs to perform deictic pointing to objects in the real world. We believe that enabling IVAs with pointing gestures can enrich the communication channel and promote efficient human-like interactions [32].

To enable IVAs to point effectively, we seek to answer how accurately users can interpret the direction of an IVA's pointing, to establish a fundamental building block for designing deictic interactions between users and IVAs. However, it remains unclear whether it is feasible to design an IVA that have users accurately recognize where the IVA is pointing into the real world. Optimally, users should be able to interpret an IVA's pointing to the real world as well as, or even better than a real person's pointing.

To explore this potential, we introduce design factors that may improve the chances that users would be able to perceive the IVA's pointing accurately to demonstrate feasibility. These include the *situated display*, *IVA appearance* and *pointing gesture* strategies. For the *situated display*, we used a spherical Fish Tank Virtual Reality (FTVR) display in our IVA design. Unlike immersive displays, the spherical FTVR display is calibrated to be viewer-aware in the real-world coordinate system, enabling the IVA to point from the virtual world to objects in the real world. It also offers effective 3D depth cues for pointing perception (i.e., stereoscopic cue and motion parallax) [33]. Besides, spherical displays have been found to provide better gaze [26,48], size and depth [61] perception compared to the flat displays. For the *IVA appearance*, we used an animated cartoon character that was not photo-realistic but offered natural, easy-to-control pointing affordances to avoid the Uncanny Valley [45] effect. For the *pointing gesture*, we designed our IVA to point following the arm vector instead of the eye-fingertip alignment commonly found in human pointing [7, 28–30, 35] as it has been shown to provide a more accurate cue.

With a real person pointing as the baseline for comparison, we conducted an empirical experiment to investigate how accurately users can perceive our IVA's pointing. As an IVA is usually smaller than a real person due to the size constraint of typical displays, we controlled for retinal size in the experimental design. Our results demonstrated that it is feasible to have an IVA accurately point to locations in the real world. Further, the IVA's pointing location was perceived as accurately as a real person in our configuration. Specifically, the IVA outperformed the real person in the vertical dimension and yielded the same level of accuracy horizontally. We also discuss how the set of design factors may have contributed to the result and suggest design implications. Thus, the design factors we suggest provide a foundation for future studies on exploring the relative importance of each factor to consider for the design of IVA with pointing gestures. We believe our study and IVA design help pave the way for research on users' perception of pointing either in the virtual environment or in the real world.

## 2 RELATED WORK

### 2.1 Pointing in Intelligent Virtual Agents (IVAs)

Pointing is a fundamental building block of human communication [34]. The ubiquity of pointing drives research on incorporating it for intelligent virtual agents (IVAs) in virtual environments [51].

Most prior studies on IVAs with pointing focus on its benefits in drawing users' attention to some content in the virtual world where the IVA is situated. For example, the Persona agent [2] could point to images on web pages and Jack, as a virtual meteorologist, can give a weather report by pointing to the weather images [47]. Atkinson [4] showed an animated virtual agent serving as a tutor in a knowledge-based learning environment and demonstrated the benefits of pointing in directing the learners' attention. When combined with speech and context, using the Behavior Expression Animation Toolkit (BEAT), an agent was created to generate correlated gestures by extracting the linguistic and contextual information from the input text [13]. To achieve deictic believability, Lester et al. [37] designed COSMO agent, using a deictic planner to determine the generation of pointing guided by the spatial deixis framework. Rather than pointing to the virtual environment, an agent called *MACK*, in mixed reality, could point to a physical paper map shared with users in reality along with speech [12]. However, an unanswered question is how accurately can an IVA point to the real world.

### 2.2 Perception of Pointing in the Real World

When humans perform pointing naturally, without instructions, instead of pointing using their arm vector, Bangerter & Oppenheimer [7] and Henriques & Crawford [28] observed that humans commonly orient their arm so that the fingertip intersects the line

joining the target and their dominant eye while gazing at the target. This is called *eye-fingertip alignment* as illustrated in Figure 2 c-2. This mechanism was further shown in the estimation of human pointing direction. With various methods proposed, the head-hand line [16, 39, 44] (also known as the eye-fingertip line) was found to be the most reliable (90%) compared to forearm direction and head orientation [46]. Mayer et al. [41] demonstrated that it yielded the lowest offset among four other ray cast techniques. As our study is to find factors that enable an IVA to point into the real world accurately, the impact of different alignment strategies is considered in our IVA design.

Pointing behavior during interpersonal interaction typically involves the movement of eye gaze, head and arm [28]. Considerable research has been done targeting gaze perception. People can accurately discern their mutual gaze with another person [3, 24] and the direction of the other person's gaze [23]. By contrast, research on the perception of pointing accuracy is scant. By evaluating the detection accuracy for different combinations of head, eye and hand pointing cues, Butterworth and Itakura [10] showed that pointing can improve spatial localization of targets when compared to head and gaze cues but suggested that pointing had limited accuracy. Bangerter & Oppenheimer [7] contested their findings with a more precise measurement technique. The results revealed that the detection accuracy was comparable to the accuracy level for eye gaze and it was unaffected by the exclusion of eye gaze and head orientation. Despite the good accuracy, they observed a perceptual bias towards the side of the pointer's arm away from the observer in the horizontal dimension and above the target in the vertical direction. It was illustrated that the ambiguity introduced by the deviation between the eye-fingertip line and arm line might account for it. A study by Cooney et al. [19] evaluated the pointing accuracy in the horizontal direction and replicated Bangerter & Oppenheimer's results. Considering the ambiguity shown in human pointing and exploiting the fact that we have explicit control over the IVA's head, eye and finger positioning, we designed our IVA to use arm vector pointing rather than eye-fingertip alignment as an approach to improve its pointing accuracy as illustrated in Figure 2.

Finally, during interpersonal interactions, the accuracy with which observers can detect the pointed targets based on another person's pointing gestures has been a key issue. Because if a person cannot accurately interpret the other's pointing direction, it would be difficult to establish joint attention within a conversation [10]. Prior research shows that the distance between users and targets can affect users' interpretation of the pointing direction [5, 16, 59]. To study this effect, we configured the distance as an independent variable to investigate how the accuracy changes in different distances.

### 2.3 Perception of Pointing in Virtual Environments

While pointing is ubiquitous in daily interactions within the real world, it is difficult for users to precisely interpret the pointing direction in virtual environments. Wong and Gutwin [59] compared users' accuracy in a collaborative virtual environment (CVE) with the real world and observed worse performance in CVE although the difference was smaller than expected. The immersive head-mounted displays (HMDs) and virtual reality (VR) systems (e.g., CAVE) only support pointing within the virtual environment where the IVA is situated. By merging the real world with the virtual environment, FTVR [58] displays enable the IVA to point from the virtual world to the real world and provide a mixed reality experience. Our experiment used a spherical FTVR display because it has advantages over other VR/AR displays and planar displays as we will discuss in Section 3.1.

Regarding the evaluation of users' perception of pointing in FTVR, previous research focused on the assessment of pointing cues. Kim et al. [33] classified the pointing cue factors to consist of three levels: gaze, hand, and gaze+hand. They found no significant

difference among the three levels with an experiment in a cylindrical 3D display. Using gaze to convey pointing direction within a spherical display has also been shown to be effective [25, 33, 48].

The research listed above is mostly concerned with telepresence. That is, the remote person is represented by an avatar or captured using cameras to realize remote collaboration. By contrast, we are using the IVA to perform pointing. In this context, the IVA is regarded as a social entity to mimic human intelligence [32] and work with a person. Unlike pointing in telepresence, designing the IVA's pointing gestures provides more possibilities to improve users' perception of pointing as the pointing behaviours do not have to be exactly human-like. Thus, for our design, we have the opportunity to design the IVA's pointing gestures not completely the same as humans. This enables us to remove the eye-fingertip alignment in the IVA as suggested in Section 2.2. The complete IVA design is discussed in the following Section 3.

## 3 DESIGN FACTORS

This section elaborates on the design factors to enable our IVA to point as accurately as possible, including the *situated display*, *IVA appearance* and *pointing gesture* strategies.

### 3.1 Situated Display

We used a spherical FTVR display for IVA due to the following reasons. First, FTVR displays are situated in the real world which enables the IVA to point from the virtual environment to locations in the real world. Alternative approaches, such as immersive headset displays, only support pointing within the virtual environment where the IVA is situated. Though AR displays provide the see-through feature that can get similar effects, these systems lack the tangible nature of having a volumetric display that is part of the real world. FTVR displays also provide motion parallax and stereoscopic cues, which are important in interpreting pointing gestures [33]. The spherical shape has been found to provide better depth and size perception compared to a planar counterpart [61]. Spherical screens also showed better task performance in perceiving gaze direction compared to planar screens [26, 48]. As perceiving pointing gestures depends on multiple aspects of visual perception such as depth and orientation perception, it is promising to use spherical FTVR displays to improve the pointing perception.

### 3.2 IVA Appearance

The state of the art in photo-realistic representations for IVAs is subject to the Uncanny Valley [45]: a high degree of realism does not necessarily lead to positive evaluations. Considering this effect, Schneider et al. [54] suggest to use a non-human appearance with the ability to behave like a human. Following this suggestion, we chose a Japanese female cartoon character as our IVA to avoid the negative feelings caused by a human-like appearance while supporting human-like behaviors. Our IVA's appearance is designed with large eyes and small nose (Figure 3) to provide the characteristic of the baby schema [40], which could induce a pleasurable feeling [27].

### 3.3 Pointing Gestures

We designed our IVA to point following the arm vector (Figure 2 c-1) instead of the eye-fingertip alignment (Figure 2 c-2) to avoid potential perceptual ambiguity. As discussed in Section 2.2, humans commonly point to where they are looking by aligning their fingertip with the gaze of their dominant eye [7, 28] (Figure 2 c-2). When it comes to perceiving others' pointing, this might introduce ambiguity because the location followed by the arm vector is different from the actual target location followed by the eye-fingertip line. Previous work [7] found that participants exhibited a perceptual bias towards the upside of the target, potentially because of this ambiguity. Therefore, rather than design IVAs to point the same way as humans commonly do (i.e., eye-fingertip alignment), we instead remove the

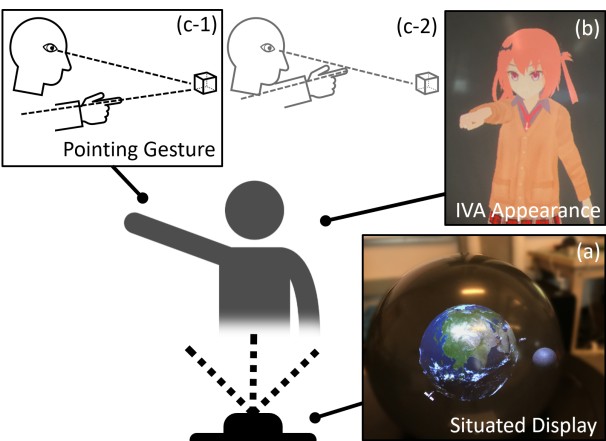

Figure 2: We consider three design factors to enable accurate perception of the pointing performed by IVA, including situated display, IVA appearance, and pointing gesture. (a) We used a situated spherical 3D display as it offers effective depth cues for pointing perception. (b) We used an animated cartoon character that offers natural, easy-to-control pointing affordance. (c) We designed our IVA to point following the arm vector (c-1) instead of the eye-fingertip alignment (c-2) to avoid potential perceptual ambiguity.

eye-fingertip alignment in the design of IVA's pointing gestures, that is, the arm vector directly points at targets (Figure 2 c-1). Our expectation is that this approach will mitigate the perceptual errors of eye-fingertip alignment and result in a perceptually accurate IVA pointing gesture.

For pointing cues, previous research has found that the orientation of the pointer's eyes, head and hand are used as visual cues for an observer to interpret a pointing gesture [28, 33]. Prior work [60] has found that the hand cue alone provides accurate pointing perception but with a loss of naturalness. In our study, we decide to include all the pointing cues, i.e., eyes, head and hand orientations, to promote accurate and natural perception. In summary, we design our IVA to point with an outstretched arm, eyes and head facing the target without the eye-fingertip alignment; thus, all cues are consistently directing attention to the same location.

## 4 EXPERIMENT

The goal of our experiment is to assess how accurately our IVA can point into the real world. With a real person's natural pointing as the baseline, we measured how accurately a human observer can interpret the pointing of our IVA or the real person to a physical location. In doing so, we lay the foundation for the design of IVAs with pointing and shed light on future studies about the contribution of the individual design factor.

### 4.1 Participants

Thirty-six participants (19 females and 17 males) aged between 21 and 30 were recruited from a local university to participate in the study with compensation of a $10 gift card. All had normal or corrected-to-normal vision.

### 4.2 Apparatus

We set up the experiment using a situated 24-inch spherical display (Figure 3) which renders the IVA, and a flat fabric projector screen which renders target area. With four stereo projectors rear projecting onto the spherical surface, we adopted an automated camera-based multi-projector calibration technique [63] to enable a 360-degree seamless image with 1-2 millimeter accuracy. The projectors are

Optoma ML750ST with $1024 \times 768$ pixel resolution and a frame rate of 120 Hz. With an NVIDIA Quadro K5200 graphics card, a host computer sends rendering content to the projectors. Our IVA was rendered using Unity3D and MikuMikuDance [31] model from DeviantArt [50]. We used an OptiTrack™ to track the passive markers attached to the shutter glasses for head tracking. We used a pattern-based viewpoint calibration [56] that computed a viewpoint registration with an average angular error of less than one degree. Viewers gain perspective-corrected images with stereo rendering coupled with the synchronized shutter glasses. The total latency is between 10-20 msec [22]. With a resolution of $34.58ppi$, the display provides various 3D depth cues such as motion parallax and stereoscopic cues [62]. An Optoma ML750ST projector with $1024 \times 768$ pixel resolution was used to display an $80cm \times 80cm$ target area on a flat fabric projector screen. The grid content and target indicator were created by Unity3D.

## 4.3 Human and IVA Pointing

As a baseline, an independent real person (RP) was hired to be the pointer. The dominant hand and eye of our RP are both on the right side. To capture the specific natural human pointing as the baseline, RP was not instructed about the specific manner about pointing gestures but just asked to point as accurately as possible with head, eyes and outstretched arm. Both RP and IVA used the left arm to point to the targets in the left region and the right arm for targets in the center or right region.

In practice, most IVAs are rendered in relatively small displays such as home assistant systems [1, 11, 20]. The size difference between IVA and RP makes it challenging to make a fair comparison on the pointing perception. To characterize the potential effect of the size difference, we include two viewing conditions in our study: SameDis and SameRet (Figure 4). In SameDis, the IVA and RP are placed at the same observation distance from the participant. In this condition, the retina image of IVA is smaller than RP with the arm length of IVA and RP as $30.5cm$ and $68cm$ respectively. In SameRet, the retina sizes of IVA and RP are the same by moving RP $56cm$ further away from the participant, resulting in the same angular size of the arm length in IVA and RP. The viewing condition is designed based on previous study that found the task performance of visual reasoning did not vary as long as the retinal image is unchanged, demonstrated through an experiment with a larger display placed farther than a smaller display [14]. However, moving RP away may introduce potential experimental bias by increasing the viewing distance. We included both conditions (same retinal size & same viewing distance) to see what impact the size factor has.

## 4.4 Experimental Design

We followed a $2 \times 2 \times 2$ mixed design with one between-subjects variable (**C1**) and two within-subjects variables (**C2, C3**):

- **C1** The *Viewing* condition, which could be Same Retinal Size (SameRet) or Same Distance (SameDis). In SameDis, the viewing distances in RP and IVA are the same. In SameRet, the retinal sizes are the same by placing RP $56$ *cm* farther from the participant compared to IVA (Figure 4).

- **C2** *Pointer* which could be Intelligent Virtual Agent (IVA) or Real Person (RP).

- **C3** *Distance* which could be near or far. The distance between the participant and target area is 70 *cm* in near and 210 *cm* in far.

We designed **C1** as a between-subjects variable to avoid learning and transfer across different viewing conditions. We randomly and equally divided 36 participants into 2 groups. One group went through SameRet and the other did in SameDis. Each group went through the levels of **C2** $\times$ **C3**. The order of **C2** and **C3** was fully counterbalanced.

We measured error and error bias in the horizontal and vertical dimensions, suggested by prior study that has found systematic bias particularly in the vertical direction [7, 19]. We collected subjective data through a post-study interview. The quantitative metrics are as follows:

- *Horizontal & Vertical Error*, defined as the Euclidean distance between the actual target location and participants' perceived location along the corresponding axis.

- *Horizontal & Vertical Error Bias*, computed by subtracting the actual position from the perceived location. A positive value indicates an estimation to the right or above the true locations, respectively.

## 4.5 Task

In each trial, participants observed the pointing performed by IVA or RP and reported the pointing position by clicking where they believe the IVA or RP was pointing using a mouse. They were asked to prioritize accuracy over speed. The pointing positions are located within an $80cm \times 80cm$ square projected onto a fabric projector screen as the target area placed beside participants (Figure 4). Early pilot of this task has shown that the task might be too difficult due to the lack of reference in a blank background. Therefore we provided a relatively dense $40 \times 40$ line grid as the target background (Figure 4).

## 4.6 Procedure

Participants started by filling out a consent form followed by verbal explanations of the experiment. Participants were asked to sit on an adjustable chair (Figure 3) to ensure the horizontal alignment of their shoulder and the pointer's shoulder in both IVA and RP. They were seated by the right side of the pointer (Figure 4). The distance between the participant and the target area is 70 cm in near, and 210 cm in far, which are chosen to represent the proximal pointing in the near distance and approximate the distal pointing [53] constrained by the experimental room.

Each participant was provided with a mouse and a clipboard to hold it. They were instructed to click where they believe the IVA or RP was pointing by prioritizing accuracy over speed. With a total of 4 conditions (IVA vs RP, Near vs Far), each contains 20 trials at different locations with the first 5 provided as practice located at left middle, right middle, top middle, bottom middle and center to illustrate the entire region. Participants were told the actual location in the practice trials. In the formal trials, the locations of targets were randomly generated and can be any location inside the target area. To avoid cross-talk with previous targets serving as a reference, participants were instructed to close their eyes between trials.

When the participant was ready, the IVA pointed to random locations inside the target area, controlled by the experimenter using a keyboard, whereas RP performed the pointing gesture using a visible random target while the participant had their eyes closed. The dominant hand and eye of our RP are both on the right side. Both IVA and RP used the left arm to point to the targets located in the left region and the right arm for targets in the center or right region. When the gesture was ready, the reference target for RP disappeared and participants were asked to open their eyes to perform the task. The IVA and RP held the gesture until the participant had finished the click and said "okay." No other communication was provided between participants and RP. Once all conditions were completed, a follow-up interview was conducted to collect participants' subjective preference between IVA and RP on the easiness to perceive pointing and the difference between the perceived and actual pointed location. We also asked the pointing cues that they referred to in the task. The study took approximately 30-40 min to complete.

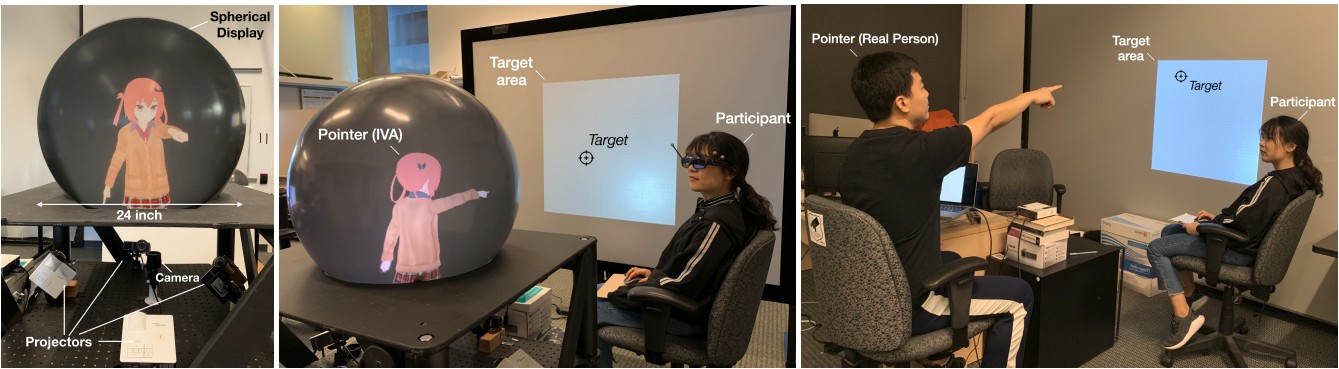

Figure 3: (Left) The Intelligent Virtual Agent (IVA) in a spherical Fish Tank Virtual Reality (FTVR) display that enables the IVA to point from the virtual world to the real world. (Middle) Experimental setup with IVA as the pointer. A participant wears tracked shutter-glasses to perceive the perspective-corrected stereoscopic IVA on the spherical FTVR display. (Right) Experimental setup with a real person (RP) as the pointer. A RP was hired to perform natural pointing as a baseline for the comparison with the IVA's pointing.

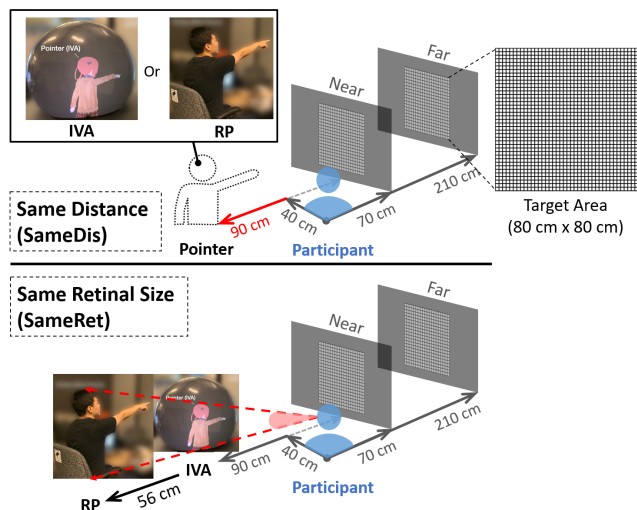

Figure 4: Experimental layout in the viewing conditions. (Top) Same Distance (SameDis): Real person (RP) and IVA have the same viewing distance with respect to the participant. The retina size of IVA is smaller than RP. (Bottom) Same Retinal Size (SameRet): RP and IVA have different viewing distances with respect to the participant to keep the same retinal size.

tal error in near *Distance* ($M = 9.98cm, SE = 0.35cm$) was 26.5% lower (***) than in far ($M = 13.58cm, SE = 0.52cm$). We did not find main effects of *Viewing* ($F(1,34) = 0.44, p > 0.05$) and *Pointer* ($F(1,34) = 0.44, p > 0.05$). No interaction effects were found among factors.

***Vertical Error***: We found main effects of *Pointer* ($F(1,34) = 29.42, p < 0.001$) and *Distance* ($F(1,34) = 31.74, p < 0.001$) for the vertical error. The mean vertical error in IVA ($M = 10.22cm, SE = 0.37cm$) was 28.8% lower (***) than in RP ($M = 14.35cm, SE = 0.56cm$). The mean vertical error in near *Distance* ($M = 11.11cm, SE = 0.47cm$) was 17.5% lower (***) than in far ($M = 13.46cm, SE = 0.52cm$). We did not find main effect of *Viewing* ($F(1,34) = 2.66, p > 0.05$). A two-way interaction effect was observed between *Viewing* and *Pointer* ($F(1,34) = 5.05, p < 0.05$).

A post-hoc analysis of the two-way interaction effect *Viewing* × *Pointer* (Figure 5(d)) shows significant difference in vertical error between RP and IVA in both SameRet ($p < 0.05$) and SameDis ($p < 0.001$). When viewing in SameRet, the mean vertical error in IVA ($M = 10.47cm, SE = 0.48cm$) was 18.7% lower (*) than in RP ($M = 12.88cm, SE = 0.77cm$). When viewing in SameDis, the mean vertical error in IVA ($M = 9.98cm, SE = 0.57cm$) was 36.9% lower (***) than in RP ($M = 15.81cm, SE = 0.75cm$). The mean vertical error was significantly lower (*) in SameRet ($M = 12.88cm, SE = 0.77cm$) than in SameDis ($M = 15.81cm, SE = 0.75cm$) in RP ($p < 0.05$), but not ($p > 0.05$) in IVA.

### 4.7 Data Analysis

We conducted a mixed ANOVA with **C1** *Viewing* as a between-subjects factor, **C2** *Pointer* and as **C3** *Distance* as within-subjects factors. Significance values were reported in brackets for $p < .05(*), p < .01(**)$, and $p < .001(***)$ respectively. Numbers in brackets indicate mean ($M$) and standard error ($SE$) for each respective measurement. The post-hoc analysis was conducted using pairwise t-tests with Bonferroni corrections.

### 4.8 Results

#### 4.8.1 Error

With all assumptions met, we used a $2 \times 2 \times 2$ mixed-model ANOVA (*Viewing* × *Pointer* × *Distance*) on the Horizontal Error and Vertical Error respectively (Figure 5(a)).

***Horizontal Error***: We found main effect of *Distance* for the horizontal error ($F(1,34) = 69.16, p < 0.001$). The mean horizon-

#### 4.8.2 Error Bias

With all assumptions met, a mixed-model ANOVA was conducted on the Horizontal and Vertical Error Bias respectively. The means and 95% CIs and a scatter plot showing the error bias for all participants can be found in Figure 5(b)(c).

***Horizontal Error Bias***: We did not find main effects of *Pointer* ($F(1,34) = 3.17, p > 0.05$), *Distance* ($F(1,34) = 2.47, p > 0.05$), *Viewing* ($F(1,34) = 0.14, p > 0.05$) for the horizontal error bias, or any interaction effects among three factors.

***Vertical Error Bias***: We found main effect of *Pointer* ($F(1,34) = 284.84, p < 0.001$) for the vertical error bias. The mean vertical error bias in IVA ($M = -2.41cm, SE = 0.73cm$) was significantly lower ($p < 0.001$) than in RP ($M = 13.47cm, SE = 0.64cm$). We did not find main effects of *Distance* ($F(1,34) = 1.66, p > 0.05$), *Viewing* ($F(1,34) = 2.20, p > 0.05$), or any interaction effects among three factors.

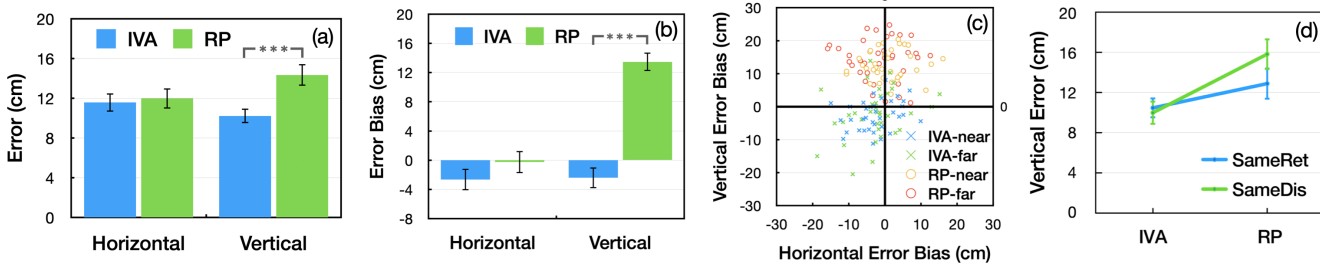

Figure 5: Study results on (a) error, (b-c) error bias, and (d) interaction effect *Viewing × Pointer* on the vertical error. (a) Mean error and 95% CIs of perceived pointing locations in IVA and RP. IVA yielded significantly lower error (28.8% less) in vertical dimension and comparable horizontal accuracy as RP. (b) Mean error bias and 95% CIs in IVA and RP. Participants showed a systematic upward bias in perceiving RP's pointing, which is demonstrated in (c) the scatter plot of all participants' average error bias. Data points above the horizontal axis indicate upward bias. Significance values were reported for $p < 0.05(*), p < 0.01(**)$, and $p < 0.001(***)$.

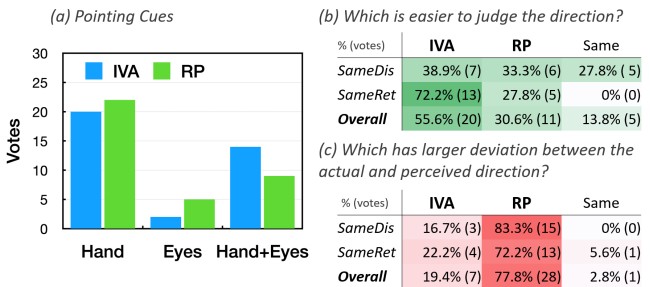

**(b) Which is easier to judge the direction?**

| % (votes) | IVA | RP | Same |
|---|---|---|---|
| *SameDis* | 38.9% (7) | 33.3% (6) | 27.8% ( 5) |
| *SameRet* | 72.2% (13) | 27.8% (5) | 0% (0) |
| *Overall* | 55.6% (20) | 30.6% (11) | 13.8% (5) |

**(c) Which has larger deviation between the actual and perceived direction?**

| % (votes) | IVA | RP | Same |
|---|---|---|---|
| *SameDis* | 16.7% (3) | 83.3% (15) | 0% (0) |
| *SameRet* | 22.2% (4) | 72.2% (13) | 5.6% (1) |
| *Overall* | 19.4% (7) | 77.8% (28) | 2.8% (1) |

Figure 6: Participants' preference between IVA and RP for (a) pointing cues, (b) easiness of judging the pointing, and (c) deviation between the actual and perceived pointing direction. (a) Most participants took Hand as the major cue in determining the pointing direction of both IVA (55.6%, 20/36 participants) and RP (61.1%, 22/36). (b) The IVA was selected as the easier one to perceive pointing with (55.6%, 20/36) especially in SameRet condition (72.2%, 13/18). (c) Most participants (77.8%, 28/36) found RP exhibited a larger deviation between the actual and perceived pointing in the practice trials than IVA.

### 4.8.3   Interview Responses

In the post-study interview, we found that most participants took Hand as the major cue in determining the pointing direction of both IVA (55.6%) and RP (61.1%) as shown in Figure 6(a). We collected participants' subjective preference between IVA and RP on the ease with which to perceive pointing (Figure 6(b)). We found that a large majority of participants (72.2%) chose the IVA in the SameRet condition as the easier one. Since participants were told the actual locations pointed to in the practice trials before each session, we collected subjective data about whether their expected locations are close to the actual ones to understand their perception of pointing (Figure 6(c)). 77.8% participants found that the difference between the actual and perceived location was larger in RP compared to IVA, with 72.2% in SameRet and 83.3% in SameDis.

## 5   DISCUSSION

Based on the results, we summarize the following major findings. We found that participants can perceive accurately where the IVA was pointing in the real world:

- IVA achieved accurate pointing perception with the horizontal error of 11.58 cm comparable to 11.99 cm of RP and the vertical error of 10.22 cm significantly lower than 14.35 cm of RP.

- Participants showed a systematic upward bias of 13.47 cm regardless of *Distance* in RP but not in IVA.

- The *Viewing* condition did not appear to affect the accuracy difference between IVA and RP.

### 5.1   Reflections on Design Factors

In this section, we discuss the three design factors (pointing gesture, situated display, and IVA appearance) to provide interpretations of our findings in relation to RP as well as suggest future directions.

#### 5.1.1   Pointing Gestures

In our study, RP was asked to point as accurately as possible by naturally moving their head, eye-gaze, and outstretched arm towards the target. After the experiment, we asked the RP about their pointing and found the RP used eye-fingertip alignment. This is not surprising, as it is commonly observed in natural human pointing [7, 28]. However, we found that participants might perceive the pointing gesture in a way different from how RP performed it. This difference between perceiving and performing the pointing gesture may potentially explain the strong upward bias observed in the RP results (Figure 5(b)).

To illustrate this bias, consider Figure 7(a): a pointer outstretches its arm to point to a target (green cube) by placing the fingertip on the line joining the dominant eye and the target. If the viewer perceives the pointing direction by following the arm vector extended from the fingertip, there will be a vertical error causing an incorrect position (blue cube) with an upward bias deviating from the actual pointed position (green cube). Note that regardless of the actual target position, the vertical error will always be positive (upwards) since the pointer's shoulder (the origin of the arm vector) is always below their eyes (the origin of eye-fingertip vector). Similarly, there will also be horizontal bias as shown in Figure 7(b). Different from the vertical bias, horizontal bias can be both positive (on the right) and negative (on the left), which could potentially explain that we only observed systematic positive error bias in the vertical direction but not in the horizontal direction in RP (Figure 5(b)). The systematic upward bias of RP (13.47 cm) is consistent with prior work [7] in which they found a mean angular error bias of 2.5 degree above the target equivalent to a mean vertical error bias of 11.26 cm averaged across the viewing distances used in our setup.

In the post-study interview, the majority of participants (61.1% in RP and 55.6% in IVA) reported that they mainly focused on the hand/arm cue as the reference to find the pointing direction (Figure 6(a)). This is consistent with prior work [38] as they found users might employ an imaginary ray extending from a fingertip to perceive the pointing in a similar referencing task. In addition,

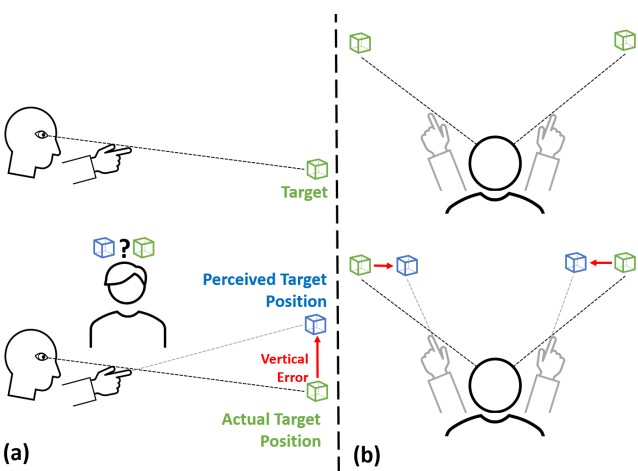

(a)           (b)

Figure 7: Illustration of the error bias when using a pointing posture with eye-fingertip alignment. A pointer outstretches the arm to point to a target (green cube) by placing the fingertip on the line joining the dominant eye and the target. (a) In the side-view, the perceived location (blue cube) is systematically higher than the actual location (green cube). (b) In the top-view, left and right arm pointing with eye-fingertip alignment results in the deviation in both directions.

77.8% (28) participants (Figure 6(c)) reported a large deviation in RP between the actual and perceived location with 16 participants commenting that it was confusing to find that RP pointed higher vertically than expected. By contrast, 19.4% reported the deviation and confusion in IVA. It indicates that the IVA's pointing gesture, which is the arm-vector pointing rather than the eye-fingertip alignment, is likely to be more aligned with how the majority perceived the pointing direction.

Besides the error bias, we also found the vertical error in IVA (10.22 cm) was significantly lower than RP (14.35 cm) as shown in Figure 5(a), also reflected in participants' interview responses on the ease of perceiving the pointing of IVA (Figure 6(b)). Without the eye-fingertip alignment, the correct target location would be reached directly by following the arm vector of IVA. Therefore, we suggest using the arm-vector as the primary cue when designing the pointing gesture of IVA for higher accuracy of pointing perception.

Our results showed that participants were more accurate horizontally than vertically (Figure 5). One potential cause is the difference between horizontal and vertical visual acuity. Previous work found [15, 17] that users have better horizontal visual acuity to perceive gaze directions compared to vertical acuity. Another possible explanation is that the arm switch of the pointer might provide a visual cue as to which side the target is located. The left/right arm inherently implies the left/right region, potentially making the task easier horizontally than vertically. Further experiments are needed to investigate it.

### 5.1.2 Situated Display

Our study was conducted using a spherical FTVR display. It is an open question as to how the findings can be applied to other display devices such as non-FTVR 2D monitors and HMDs. Conventional flat displays, like monitors without using FTVR, lack depth cues such as motion parallax and stereopsis, which are essential for pointing perception [33]. In addition, on a flat surface, the *Mona Lisa* effect, which describes a phenomenon in which a character's eyes seem to follow the user irrespective of the user's position [43], could negatively affect users' perception of pointing with eye gaze as a pointing cue.

Flat FTVR displays provide additional depth cues compared to traditional 2D displays. The major difference between a flat and spherical FTVR display is the shape factor. We expect it would be difficult to achieve similar results on a flat FTVR display due to two reasons. First, existing studies have found that spherical FTVR displays provided better gaze, depth and size perception than a flat counterpart [26, 61]. Perceiving pointing direction depends on the depth and size perception. Another related issue is the vergence-accommodation conflict (VAC) [57]. Although participants in our experiments were required to sit on a fixed chair, we did not constrain their head motion. With a spherical display, they could keep a relatively constant screen distance following a curvature [61]. While for the flat counterpart, users' viewing distance to the screen surface would change while moving their head, which might result in a more pronounced VAC. Future studies are required to investigate these issues and evaluate IVAs' pointing accuracy within the flat FTVR displays.

For other 3D displays with perspective-corrected and stereo rendering, such as a CAVE and HMDs used in AR, we anticipate that similar results may be found depending on the relative importance of each of the other design factors we investigated, i.e. IVA appearance and pointing gesture type. Future studies of controlled experiments would be required to understand the effect of individual factors and potentially associate the result with the display factor.

### 5.1.3 IVA Appearance

The different appearances between RP and IVA, such as gender, realism and eyes may have some influence on participants' perception. Prior research on user preferences for agents' gender presents contradictory findings and trends, which may be due to user characteristics or context [49]. Regarding realism, RP was reported by four participants to be more familiar and common. Two participants commented that IVA's bigger eyes were helpful to judge the direction. In contrast, RP's eye gaze cue was reported to be subtle by three participants, with one indicating it was even harder to discern the change in the horizontal direction. Moreover, two participants said that they tried to avoid eye contact in RP, while there was no such concern in IVA. Besides, previous research showed that users exposed to images of animals with baby schema were more physically tender in their motor behavior and performed better on a task that demanded extreme carefulness [55]. The baby schema of the IVA might have some effect on participants' performance. Future studies could determine the extent to which each aspect contributes to the pointing perception.

### 5.2 Distance

Not surprisingly, participants perceived pointing more accurately when targets were closer than farther, no matter whether it is in SameRet or SameDis. With the same target area, farther distance results in a subtler angular change for all the pointing cues (head, eyes and hand). Three participants also commented that it was hard to extend the arm line to locate the target when farther away. However, despite the higher level of difficulty for farther distances, our IVA can still point more accurately than the real person, indicating the effectiveness of our IVA design. It also suggested that users are able to know where an IVA is pointing within a range of distance. Practically, this implies that should an IVA be used as a home assistant or a virtual tutor, it can be situated in a single location and still be able to point to near and far objects while indicating, "It's over there." to provide deictic indications with users.

### 5.3 Viewing Condition

We introduced the *Viewing* condition as a between-subjects factor due to the size difference between IVA and RP. Our main finding that IVA provides better pointing perception than RP holds both in

SameRet and SameDis. Therefore, incorporating the *Viewing* condition in the study design helps to validate our results. Besides, we found the *Viewing* condition plays a role in the pointing perception with the interaction effect *Viewing* × *Pointer* (Figure 5(d)). Adjusting the distance between RP and the participant leads to different retinal sizes and causes the difference in the vertical error between SameRet and SameDis in RP. Note that the difference is only in the vertical error but not in the vertical error bias, indicating that changing the *Viewing* condition did not introduce systematic bias but affected the precision of the pointing perception. One possible explanation is that there might exist an optimal viewing distance and retinal size to perceive the pointing direction by observing the pointer's posture. Our study focused on the difference between pointers and evaluated one fixed distance or retinal size. Future studies are needed to investigate the potential effect of different viewing distances and retinal sizes on the pointing perception.

Participants' comments on the size difference are quite divergent. In SameDis where the retinal size of IVA is approximately half the size of RP, 5 out of 18 participants commented that IVA's pointing was easier. They explained that the smaller size of IVA allowed them to perceive a more noticeable change of eyes, hand and head orientation. Conversely, 4 out of 18 participants who found RP easier commented that RP's life-size was more natural to perceive the pointing. Similarly in SameRet their comments are also divergent. Four participants preferred IVA's smaller size whereas two preferred the life-size of RP. While future studies can quantify individuals' sensitivity to this factor, we also note that from a practical perspective, our study shows that there is unlikely a one-size-fits-all solution to optimize the size and visual representation of an IVA. Thus, allowing users to tailor their IVA's appearance would be advisable.

## 6 DESIGN IMPLICATIONS

The main design implication from our study is that with a set of design factors determined, it is feasible to have an IVA point with comparable accuracy to a real person. In our IVA design, we used a spherical FTVR display, rendered a 3D cartoon IVA with human-like behaviors and applied arm vector pointing instead of the eye-fingertip alignment, which collectively contributes to our IVA's high pointing accuracy. As the appearance and pointing gesture strategy are not dependent on the display factor, we expect these design choices could be considered in other display devices. The findings serve as a foundation for designing an IVA to point to the physical world accurately and provide pathways for future studies to precisely quantify the relative contribution of each factor.

We also suggest to provide more cues for perceiving pointing to objects farther away. According to our results, when participants were farther from the target, the accuracy of the pointing perception decreased significantly. Visual cues such as the orientation of the head, hand and eye gaze might not be sufficient to accurately indicate the target. Additional verbal cues, such as the location or feature description, should be considered to convey the pointing direction efficiently, which better resembles human pointing behaviour. For example, a combination of verbal description, i.e., "it's on the table over there", with a pointing gesture can be implemented with IVAs. A future study could investigate the natural communication mechanisms combining voice and deictic gestures.

## 7 LIMITATIONS AND FUTURE WORK

We discuss four limitations of our work along with opportunities they present for future research. First, we hired one single RP as the baseline pointer. Though we confirmed with RP that his pointing behavior followed the eye-fingertip alignment, future work could use a motion capture system to track the posture and provide some data about the accuracy objectively. Further, the RP in the experiment was instructed to point at the same location until the participant said 'OK'. Holding pointing for a long duration will have

unintentional movements, such as, hand tremor and jitter, which will impact the perception of pointing negatively. The IVA can be designed to hold its posture perfectly still, so does not suffer from unintentional movements. Though, these human movements may positively impact the feeling of naturalness, which may play a role in mitigating an uncanny valley when designing photo-realistic IVAs. Without explicit instructions, the RP pointed naturally in a way with eye-fingertip alignment commonly found in human natural pointing [7,28]. Our primary goal for the study was to establish that users can perceive where a carefully designed IVA is pointing. While using an RP baseline illustrated some potential avenues of research to quantify the differences in pointing between IVAs and real people, our study was not designed to do so. For example, natural human pointing has a range of variations of pointing gestures and strategies that are employed. We believe future studies can pursue with multiple RPs spanning a range of strategies, which would help establish the robustness of IVA pointing relative to human pointing, and define the lower and upper bounds of IVA/RP differences to provide additional insight into different design approaches for IVA pointing gestures and appearance.

Second, despite providing head tracking and depth cues, FTVR displays still have many technical and perceptual limitations, e.g., lower resolution and fewer depth cues than in reality. These constraints may affect participants' accuracy. Two participants pointed out that IVA lacked depth information (e.g., shadows and lighting). However, all of their quantitative data still suggests a higher accuracy in IVA than in the RP baseline. This indicates that our display's constraints did not appear to have a notable negative impact on participants' performance. The effect of display quality characteristics on the perception of pointing should be identified with further user studies.

Third, the design of an IVA involves many factors. In this paper, we focused on using a situated spherical FTVR display, a cartoon IVA appearance and arm-vector pointing gestures. We demonstrated that these factors were sufficient for the IVA to point with comparable accuracy to a real person. Future work will draw attention to controlled experiments for each of the design factors to demonstrate their effects and the degree of individual's sensitivity to the cues that we observed. For example, to precisely quantify the effect of the eye-fingertip alignment, we can have an IVA point with eye-fingertip alignment to compare with the current design. Through studying different pointing configurations, we can create a set of configurable IVA characters that individuals can personalize to optimize their interactions with the IVA.

Last, gesture and language are highly integrated components in interpersonal conversation [8, 21, 42]. Our study provides a foundation for designing IVAs that can point accurately to the real world. However, during a conversation, people do not rely on pointing gestures exclusively [7]. Typically, they will rely differently and flexibly on gestural or verbal means [6]. Thus, a future step will concentrate on the role of pointing gestures with verbal cues given to establish joint attention with the IVA.

## 8 CONCLUSION

In this paper, we proposed an IVA with design factors including a situated display, appearance, and pointing gesture strategy to investigate whether it is possible to have an IVA point accurately into the real world. Using a spherical FTVR display, we conducted a study to measure the IVA's pointing accuracy while comparing to a natural human pointing baseline. In the study, the IVA's pointing accuracy was determined by having participants estimate where they perceive the IVA is pointing in the real world. The participants also estimated where a real person was pointing using the same experimental setup for comparison and discussion of the different design factors.

Our results show that participants perceived the IVA's pointing into the real world with comparable accuracy to the real person.

Specifically, the IVA outperformed the real person in the vertical dimension and yielded the same level of accuracy horizontally. We discussed design factors that likely contributed to the success of the IVA pointing accuracy, and suggested directions for future studies to provide accurate pointing perception. Our results for the human pointing baseline are consistent with previous literature, showing that participants mainly focus on the pointer's hand, which leads to a bias when interpreting a real person's pointing direction. Particularly, we found participants exhibited a systematic upward bias in the vertical dimension when perceiving the human pointer, which we suspect is due to the ambiguity associated with the eye-fingertip alignment that is commonly employed by people when they point in the real world. The adjustment afforded by the IVA design to use arm vector pointing is helpful to improve IVA pointing accuracy.

As voice and visual interfaces for home assistants and other digital assistants are becoming commonly used in daily life, an embodied IVA that can provide gesture cues is expected to enable a more human-like interaction. We demonstrated that a well-designed 3D visual representation of an IVA can be endowed with the capability to point to the real world with comparable accuracy to a real person. Our work shows how an IVA rendered in a 3D display can provide effective pointing gestures, which could be used in conjunction with a voice interface for natural communication bridging the virtual and the real world.

## ACKNOWLEDGMENTS

The authors wish to thank reviewers for their constructive comments and NSERC for funding this project.

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
