# OpenReview forum: "It’s Over There: Designing an Intelligent Virtual Agent That Can Point Accurately into the Real World"
_graphicsinterface.org/Graphics_Interface/2022/Conference — GI 2022_

### Official Review · Reviewer_SEKa · 2022-04-12
**Interesting study with potential to spur additional research but lacking clarity in some sections**

**Rating:** 6
**Confidence:** 3

**Review:**

This paper presents and investigation into the interpretability of pointing by an intelligent virtual agent (IVA) embodied in a spherical 3D display. The results show promise for the design of IVAs with pointing capabilities.

In terms of strengths I found the study to be robustly designed with careful attention to potentially confounding factors and good choices for which of these to prioritize in this investigation (and which to leave for future work). The statistical analyses seem sound and the results for the most part well interpreted. I'm not well versed in the related work so I can't state for certain how novel the work is, but it seems to be offering  a novel contribution, with promising results upon which others can build.

In terms of weaknesses, I found the writing to be uneven. Some parts of the paper are well written and easy to understand, but others are not. In particular, some parts of the related work (e.g., 1st paragraph of 2.1) have substantial grammatical issues  that hinder understandability. I feel like I got the general idea in these sections, but may have missed some subtler points. There are also some ideas (e.g., the uncanny valley effect) that are introduced multiple times but never quite defined. This makes the paper less useful for a reader new to the field (such as a student) who might need a bit more grounding. Other ideas such as finger-tip alignment are spread across the paper in a way that feels a bit repetitive while also cognitively demanding. I feel there is an opportunity to improve and tighten up the writing  and flow, which would improve the usefulness of the paper.

I was additional unclear about the role of the design factors relative to the contribution of the paper. The paper states that they are 'introduced' but it is unclear where they come / how they were developed. Are the factors meant as an exhaustive list? An initial brainstorm? Was the point of the evaluation to in part validate them? If so, how do the findings support (or not) or suggest the need for additional research? The choices all seem reasonable but at the same time it's a bit unclear what the intention is with respect to them.

Finally, while I thought the presentation of the results and the discussion was overall quite strong, I felt there could be a bit more acknowledgement with respect to the single human pointer. While we have no reason to believe the person was a poor pointer, they may have been. Additionally, any IVA pointer will be much more consistent (I would imagine 100% consistent) relative to a human pointer. The discussion currently underplays/overlooks these and perhaps places a bit too much emphasis on arm-vector vs fingertip alignment as the reason for the IVA's success.

In sum, I lean towards recommending the paper for acceptance because the study does seem novel and robustly conducted, and could be a good grounding for future research. The problems with the writing do stand to hinder uptake of the results and I hope these can be improved in revision.

---

### Official Review · Reviewer_B8R6 · 2022-04-13
**This paper presents the results of a study that compares the accuracy of pointing between an agent and a real person. The authors find that pointing  was more accurate with the agent, whose arm directly pointed at the target,  as opposed to the real person, whose arm is aligned with their gaze pointing at the target.**

**Rating:** 7
**Confidence:** 4

**Review:**

Strengths:

Overall, the authors are careful in their study design and analysis. They also
present a useful result that can help designers of virtual agents intended to interact with users.
Specifically, that the arm, head, and gaze of the agent should directly point
to the target as opposed to one that mimics what humans do in real life.

- The authors use an interesting setup: a "fish tank" VR system to render the character so that it
  is embedded in the environment.
- The authors are careful in their study to control for the large difference
  in sizes between the pointing character and the pointing person.
- The authors are careful in the randomization of the conditions and also
  allow provide a tutorial to participants to ensure they understand the task

The limitations of the study is that the results may be specific to the fish
tank display. However, I would be curious to test the results in full VR or
augmented mobile reality.

There are a few small comments and questions I had reading the text:

1. In the abstract, the authors say that "It is a challenge to design an intelligent virtual agent (IVA) that can
point to the real world and have users accurately recognize where
it is pointing". Why?

2. It appears to be customary to measure the vertical and horizontal error
separately when evaluating pointing accuracy. Why not measure straight-line
(euclidean) distance for error?

3. In Sec 2.1, from the writeup I don't understand why the distinction between
IVAs and EVAs is important for this study.

4. I disagree that the IVA was preferred by the majority overall. From the
combined results, its basically 50-50. Only SameRet had a strong preference.

---

### Official Review · Reviewer_ydsg · 2022-04-14
**Well done, interesting, useful contribution**

**Rating:** 8
**Confidence:** 4

**Review:**

This paper presents a study on users' perception of pointing by a virtual agent rendered on a spherical 3D display.

The paper is very well organised and written and was interesting to read. The study design is carefully thought out and compares the virtual agent to a human pointing to objects at two different distances. The additional factor of viewing condition was a clever addition to mitigate confounding effects of the difference in size between the virtual agent and human. The execution of the study and the analyses are sound.

The findings provide some useful items including a more reliable accuracy by the virtual pointer and a horizontal bias of the human pointer. I appreciate that the analysis included some thoughtful steps such as considering the vertical and horizontal error distances separately and investigating the pointing bias. The interaction effect between the viewing condition and pointer is also given consideration in post-hoc analysis. The discussion provides interesting insights on the potential reasons behind the observed results, based in part on knowledge from previous work on human pointing behaviour.

Overall I think this work makes a valuable contribution to the community.

Additional comments:
- following current practices it is recommended to publish the exact p-values (to 3 decimal places).
- Is the 'R' symbol needed in the introduction (following 'Alexa')? This presents a distraction.
- One related work that is not mentioned:
Thammathip Piumsomboon, Gun A. Lee, Jonathon D. Hart, Barrett Ens, Robert W. Lindeman, Bruce H. Thomas, and Mark Billinghurst. 2018. Mini-Me: An Adaptive Avatar for Mixed Reality Remote Collaboration. In Proceedings of the 2018 CHI Conference on Human Factors in Computing Systems (CHI '18). Association for Computing Machinery, New York, NY, USA, Paper 46, 1–13. DOI:https://doi.org/10.1145/3173574.3173620

---

### Decision · Program_Chairs · 2022-04-17

Accept